# Resveratrol Inhibits Metabolism and Affects Blood Platelet Function in Type 2 Diabetes

**DOI:** 10.3390/nu14081633

**Published:** 2022-04-14

**Authors:** Anna Michno, Katarzyna Grużewska, Anna Ronowska, Sylwia Gul-Hinc, Marlena Zyśk, Agnieszka Jankowska-Kulawy

**Affiliations:** 1Department of Laboratory Medicine, Medical University of Gdańsk, 80-210 Gdansk, Poland; katarzyna.gruzewska@gmail.com (K.G.); anna.ronowska@gumed.edu.pl (A.R.); sylwia.gul-hinc@gumed.edu.pl (S.G.-H.); aja@gumed.edu.pl (A.J.-K.); 2Department of Molecular Medicine, Medical University of Gdańsk, 80-210 Gdansk, Poland; marlena.zysk@gumed.edu.pl

**Keywords:** resveratrol (Res), blood platelets (PLTs), diabetes, thrombus formation, aggregation, metabolism, thromboxane A_2_ (TXA_2_)

## Abstract

Chronic hyperglycemia contributes to vascular complications in diabetes. Resveratrol exerts anti-diabetic and anti-platelet action. This study aimed to evaluate the effects of resveratrol on metabolism and the function of blood platelets under static and in in vitro flow conditions in patients with type 2 diabetes. Blood obtained from 8 healthy volunteers and 10 patients with type 2 diabetes was incubated with resveratrol and perfused over collagen-coated capillaries. Isolated blood platelets were incubated with resveratrol and activated by collagen to assess platelet function, metabolism, ATP release, TXA_2_ production, lipid peroxidation, and gluthatione content. In the type 2 diabetes group, plasma glucose and fructosamine concentrations were significantly higher than in the healthy group. In in vitro studies, collagen-induced thrombi formation in the blood of diabetic patients was 33% higher than in the healthy group. Resveratrol reduced thrombi by over 50% in the blood of healthy and diabetic patients. TXA_2_ production was 47% higher in diabetic platelets than in the healthy group. Resveratrol reduced TXA_2_ release by 38% in healthy platelets and by 79% in diabetic platelets. Resveratrol also reduced the activities of enzymes responsible for glycolysis and oxidative metabolism in the platelets of both groups. These data indicate that the resveratrol-induced inhibition of platelet metabolism and TXA_2_ release may lead to a reduction of platelet function and thrombus formation in patients with type 2 diabetes. Therefore, resveratrol may be beneficial to prevent vascular complications as a future complementary treatment in aspirin-resistant diabetic patients.

## 1. Introduction

Chronic hyperglycemia contributes to adaptive pathologic modifications in metabolism and the function of the endothelium, and in blood platelets that lead to long-term complications such as retinopathy, nephropathy, diabetic neuropathy, and macrovascular problems in the course of type 2 diabetes. The mechanisms of hyperglycemia-induced vascular complications include an increase in advanced glycation end production, mobilization of endothelial and blood cells surface adhesion molecules, scavenger receptors, intracellular (Ca^2+^), increased reactive oxygen species accumulation, activation of purinergic receptors, increased activity of pyruvate dehydrogenase complex in blood platelets, or thromboxane A_2_ synthesis [1,2,3,4,5,6,7,8]. Despite several different treatment regiments, WHO projects that diabetes will be the seventh cause of death in 2030. Consequently, diabetes requires more research focused on finding new alternatives to treat the disease and prevent its long-term complications.

Resveratrol (3,5,4′-trihydroxy-trans-stilbene) is a type of natural polyphenol compound found in certain plants in response to injury. Sources of resveratrol in plants include the skin of grapes, blueberries, raspberries, mulberries, or peanuts. In plants, resveratrol functions as a phytoalexin, that protects against bacterial and fungal infections. Several studies have demonstrated the potent and beneficial effects of this red wine polyphenol in both animals and humans. Resveratrol is known to exert anti-oxidative, anti-inflammatory, neuro-protective, anti-carcinogenic, anti-obesity, vasorelaxing, cartilage-protective, or anti-platelet and cardioprotective effects [9,10,11].

Recently, resveratrol has been vastly studied in various animal models and in humans due to its anti-diabetic properties. Numerous studies on obese rodents, in rats with streptozotocin-induced diabetes, or in rats with streptozotocin–nicotinamide-induced diabetes demonstrated the anti-hyperglycemic action of resveratrol [12,13,14]. It was revealed that the administration of resveratrol to rats with diabetes, reduced plasma glucose or Glycated Hemoglobin (HbA1c) levels which were thought to result from accelerated glucose uptake via GLUT4 transporters and increased intracellular glucose transport in different cells [11,12,13,14,15,16]. There is also evidence from animal studies on streptozocin-induced diabetes in rats that resveratrol supplementation normalized glycemia and reduced proinflammatory Tumor necrosis factor (TNF-α), the Interleukin 1 (IL1b), and the Interleukin 6 (IL-6) levels which were linked with the improvement of insulin sensitivity and the mass and structure of the pancreas β-cells [11,12,13,14,15,16]. Moreover, resveratrol improved endothelial function in aortas of diabetic mice by inhibiting TNFα-induced activation of Nicotinamide Adenine Dinucleotide Phosphate oxidase (NAD(P)H oxidase) and restoring eNOS phosphorylation [16].

Additionally, resveratrol exerts pleiotropic action in humans and may have beneficial effects in patients with diabetes [16]. Recent meta-analyses showed that resveratrol supplementation, especially doses ≥ 100 mg/d, reduced fasting plasma glucose and increased insulin levels in patients with Type 2 diabetes (T2DM) [17]. Therefore, the observations confirm that resveratrol intakes, in the form of supplements, may be beneficial in patients with diabetes. Resveratrol is also known to have an anti-platelet effect both in in vitro and in vivo conditions [18,19,20,21].

However, there is very little data provided on the effect of resveratrol on thrombus formation, blood platelet function, and metabolism in type 2 diabetes.

In our studies, we tested the hypothesis that in hyperglycemia in patients with type 2 diabetes resveratrol reduces platelet function and thrombus formation by the down-regulation of platelet glucose metabolism and activation. Therefore, our studies also raise the possibility that resveratrol might be beneficial for controlling coagulation in patients with diabetes. Data presented here suggest that this may be the case.

## 2. Materials and Methods

### 2.1. Reagents and Materials

Resveratrol, reagents for enzymes, adhesion, gluthatione, Thiobarbituric acid reactive substances (TBARS) and ATP assays, 5-bromo-chloro-3-indoylphosphate, nitro blue tetrazolium, MTT (Thiazolyl blue formazan), and DiOC6 (3,3′-dihexyloxacarbocyanine iodide) were supplied by Sigma Chemicals Co. (Poznan, Poland); collagen was from Chrono-Log (Warsaw, Poland); Commassie Brillant Blue G-250 was from Bio-Rad (Munich, Germany); RGDS peptide was from Tocris Bioscience (Abingdon, UK); the thromboxan A_2_ (TXA_2_) kit was from Cloud-Clone Corp (Huston, TX, USA); and the fructosamine assay kit was from Biorbyt (Zurich, Switzerland). All other chemicals were of analytical grade. Venoject tubes used for blood collection and polyethylene tubing for flow studies were from Becton Dickinson (Warsaw, Poland). Rectangular boro capillaries for in vitro flow studies were from CM Scientific (Silsden, UK).

### 2.2. Experimental Groups and Sample Collection

Heparinized and citrated blood was collected from 8 healthy volunteers and 10 type 2 diabetic patients who attended the Diabetology Outpatient Department at University Clinical Center, Medical University of Gdansk, for scheduled check-ups. Healthy subjects were matched for age and sex. Patients with albuminuria higher than 0.03 g/day or with evident macroangiopathy were not included in the study. The general demographic and laboratory features of the diabetic patients are shown in Table 1.

All the blood volunteers were fasting before blood venipuncture and did not take aspirin, non-steroid anti-inflammatory drugs, phosphodiesterase inhibitors, or calcium channel blockers for at least 14 days before the venipuncture 20 mL of blood was collected from the volunteer in citrated or heparinized vacuum tubes between 7 to 9 a.m. The ethical protocol of the research was approved by the Regional Bioethical Commission at the Medical University of Gdańsk (permission NKE BN/381/2011).

### 2.3. Standard Laboratory Parameters

Plasma glucose, blood platelet count, and levels of HbA1c were analyzed during standard laboratory tests (Architect 8000 Analyzer, Abbott, Warsaw, Poland). Fructosamine was analyzed with a standard kit Ultrospec 3100 spectrophotometer (Amersham-Pharmacia-Biotech, Cambridge, UK) at 530 nm. HbA1c levels were assessed with a Variant II Hemoglobin Testing System (Bio-Rad Laboratories, GMBH, Munich, Germany).

### 2.4. In Vitro Thrombus Formation under Flow

Whole blood, anti-coagulated in sodium heparin, was incubated in the presence or absence of 0.25 mmol/L resveratrol (Res) for 30 min at 37 °C, stained with D_i_OC_6_ (5 µg/mL), and perfused over collagen-coated (50 µg/mL) capillaries (CM Scientific) at 1000 s^−1^ for 2 min, followed by washing in PBS in the same conditions. Thrombus formation was visualized using fluorescence microscopy (Nikon; Precoptic Co, Warsaw, Poland or IX83; Olympus, Warsaw, Poland). Data were calculated as the percentage of the area covered by thrombus (surface coverage) from eight representative fields using Image J (Java, National Institutes of Health, Bethesda, MD, USA).

### 2.5. Platelet Isolation

Blood platelets were isolated by the differential centrifugation of blood anti-coagulated in 3.20% sodium citrate at 150× *g* for 15 min in a 223e centrifuge (MPW, Warsaw, Poland). The obtained platelet-rich plasma was concentrated by centrifugation at 800× *g* for 15 min. Finally, the platelet pellets were reconstituted in a standard Tyrode’s buffer with 5 mM glucose to the final platelet concentration of 3–10 × 10^8^/mL (depending on the assay). For all experiments, isolated blood platelets were incubated in the presence or absence of 0.25 mmol/L resveratrol (Res) for 30 min at 37 °C.

To assess enzyme activities and ATP release, isolated platelets were incubated in a buffer containing 20 mmol/L sodium HEPES buffer, 1.7 mmol/L sodium phosphate buffer (pH 7.4), 140 mmol/L NaCl, 5.0 mmol/L KCl, and 5 mmol/L glucose. For assays on activated blood platelets, collagen (5 µg/mL) was added along with 0.20 mmol/L RGDS peptide to block platelet aggregation. Resveratrol at the final concentration of 0.25 mmol/L was added to the incubation media and then the reaction was started with platelet suspension (1 mg of protein) and continued for 30 min at 37 °C in a water bath with continuous shaking at 100 cycles/min. Reactions were terminated by the transfer of the platelet suspension to tubes placed on ice after centrifugation for 1 min at 12,000× *g*. The platelet pellets were used to analyze the activities of enzyme, TXA_2_ production, gluthatione content, and ATP release.

### 2.6. Enzyme Assays

For enzyme assays, cell samples were lysed and diluted to the desired protein concentration with 0.2% Triton X-100 for 15 min. Aconitase (EC 4.2.1.3), isocitrate dehydrogenase (EC 1.1.1.42), hexokinase, and glucose-6-phosphate dehydrogenase (EC 1.1.1.49) activities were measured by a direct measurement of NADP or NAD reduction, respectively. The increase of absorbance at the wavelength = 340 nm was measured in the spectrophotometer Ultrospec (Unicam) for 10 min at 37 °C. The activities of the enzymes were expressed in mmols of reduced NADP or NAD on the base of the absorbance ratio 6.22/mol/cm [22,23,24,25,26].

### 2.7. Cell Viability Assay

The 3-(4,5-dimethylthiazol-2-yl)-2,5-diphenyltetrazolium bromide (MTT) reduction assay was conducted to measure the activities of mitochondrial enzymes of energy metabolism (an index of cell growth and cell death) [27]. The platelets diluted in PBS were seeded in duplicate at a density of 2 × 10^7^ cells in a 96-well plate (Corning Incorporated, Warsaw, Poland). MTT was added at the concentration of 0.60 mmol/L. The incubation was continued for the next 4 h in the dark at 37 °C. Following this, DMF/SDS solution (20%/3%, pH 4.8) was added to dissolve the reduced formazan crystals of formazan. Reduced chromophore was determined by the spectrophotometric measurement at 570/690 nm (VICTOR 1420 Multilabel Counter, PerkinElmer, Warsaw, Poland). Cell viability was calculated and expressed as a percentage of the viability of control cells (100%) based on the mean absorbance values.

The fractional (S)-lactate, NAD + oxidoreductase (lactate dehydrogenase, LDH, EC 1.1.1.27), release test was used to determine the relative number of necrotic, non-living cells. The enzyme was released by non-living cells to the culture media. It was surveyed by the direct measurement of NADH oxidation at 340 mm. LDH data were expressed as a percentage of the total LDH released from the cells compared with the positive control. The positive control for the LDH assay which showed the maximal LDH release were blood cell pellets obtained by platelet samples centrifugation and homogenized in Triton X-100 at a final concentration of 0.2% by volume for 15 min.

### 2.8. ATP

For ATP assessment, the platelet pellet was deproteinized by the addition of 4%HClO_4_ and harvested into plastic tubes in ice. Protein was removed by centrifugation for 60 s at 5000× *g* and clear supernatant after neutralization with K_2_CO_3_ was used for ATP determination by the luminometric method using a Berthold Junior LB 9509 luminometer (Berthold Technology. Bad-Wilbad, Germany) [28]. The protein pellet was dissolved in 0.2 mol/L KOH and quantitated by Bradford’s assay (Bradford 1976).

### 2.9. Platelet Aggregation and Thiobarbituric Acid Reactive Species (TBARS) Assays

The platelets were suspended in 0.3 mL of medium containing Tyrode’s buffer (pH 7.4) and 5 mmol/L glucose, to obtain a density of 2–3 × 10^8^/mL, and preincubated for 5 min at 37 °C in an APACT aggregometer (Labor, Ahrensburg, Germany). The aggregation was activated by the addition of collagen (5 µg/mL) and continued for 10 min. The reaction was terminated with 10% trichloroacetic acid, and the accumulation of TBARS was assessed as described elsewhere [28,29].

### 2.10. Platelet Adhesion

The platelets were diluted in Tyrode’s buffer (pH 7.4) to obtain a density of 1 × 10^8^ mL and preincubated for 20 min at 37 °C with 0.20 mmol/L RGDS peptide to prevent aggregation. Next, the platelets were incubated with resveratrol and plated in duplicate in 96-well plates precoated with collagen (5 µg/mL) and blocked with 5% FBS (Fetal Bovine Serum). After 30 min of incubation at 37 °C, the plates were washed three times with PBS to remove any non-adherent platelets. Platelet adhesion was assessed by a spectrophotometric method using an acid phosphatase method and analyzed in a VICTOR 1420 Multilabel Counter, PerkinElmer (Warsaw, Poland). The results were assessed as percentage compared with the controls which were 100%.

### 2.11. Platelet Thromboxane A_2_ (TXA_2_) Synthesis

Platelet supernatants were stored at −80 °C, thawed, and assayed on the Cloud-Clone Corp (Huston, TX, USA), as directed by the manufacturer, at 450 nm in a microplate spectrophotometer VICTOR 1420 Multilabel Counter (PerkinElmer, Warsaw, Poland).

### 2.12. Gluthatione Assay

Briefly, the assay is based on the direct reduction of DTNB to formazan [30].

Platelet samples were deproteinized in 5% sulfosalicylic acid and centrifuged at 3500× *g* for 3 min at 4 °C. The obtained supernatant was transferred to fresh tubes. To determine total gluthatione content, the samples were placed in a reaction buffer containing 95 mM K_2_HPO_4_, 0.95 mM EDTA, 0.031 mg/mL DTNB, and 0.115 U gluthatione reductase. For samples where oxidized gluthatione was determined, gluthatione reductase was not added. The reaction was started with 50 µM NADPH and the absorbance was assessed in a VICTOR 1420 Multilabel Counter, Perkin Elmer spectrophotometer (Warsaw, Poland), at 405 mM.

Quantification of gluthatione levels was based on the reduced form of gluthatione (GSH). The calibration curve was performed with 0–500 µM standard GSH solution. The reduced gluthatione content was calculated as the difference between the total gluthatione and its oxidized form.

### 2.13. Protein Assay

Platelet protein was analyzed according to the protocol of Bradford with bovine immunoglobulin as a standard [31].

### 2.14. Statisitcal Analysis

Data are presented as the means ± SD from 8 to 10 observations. The data distribution was tested by the Kolmogorov–Smirnov normality test. Values of *p* < 0.05 were considered statistically significant. All statistical analyses were performed using the Graph Pad Prism 5.0 statistical package (Graph Pad Software, San Diego, CA, USA).

## 3. Results

### 3.1. Characterisation of the Experimental Groups

Plasma glucose and fructosamine concentrations in type 2 diabetic patients were 40% and 39%, respectively, higher compared with those in healthy people (Table 1). HbA1c levels in healthy people were below the recommended cut-off value for non-diabetic people, and in the patients with diabetes were above the general diagnostic criteria according to the Polish Diabetes Association and the American Diabetes Association (Table 1).

### 3.2. Effect of Resveratrol on Energy Metabolism in Blood Platelets of Healthy People and Patients with Diabetes

The activities of cytosolic enzymes responsible for glucose metabolism, hexokinase and glucose-6-phosphate dehydrogenase, in diabetic platelets were found to be 32% and 25%, respectively, higher than in the platelets from healthy people (Table 2). Resveratrol (0.25 mmol/L) reduced hexokinase activity by 49% and 30%, respectively, in platelets from healthy people and patients with diabetes (Table 2). Resveratrol reduced platelet G-6-P dehydrogenase activity by approximately 50% in healthy people and by 62% in patients with diabetes (Table 2).

The activity of aconitase was 24% higher in platelets from diabetic patients than in the healthy ones (Table 2). Resveratrol reduced aconitase activity by 49% and 81%, respectively, in the platelets of healthy people and patients with diabetes (Table 2). At the same time, hyperglycemia in diabetes caused no alterations in the activity of isocitrate dehydrogenase (Table 2). However, resveratrol significantly reduced the activity of the mitochondrial enzyme in both groups, and the reduction was significantly higher in the platelets of patients with diabetes (Table 2). There was no alteration in the release of ATP in collagen-activated blood platelets in diabetes compared with healthy donors (Table 2). Resveratrol reduced collagen-induced ATP release by 64% and 54%, respectively, from the platelets of healthy people and patients with diabetes (Table 2).

### 3.3. Effect of Reseveratrol on Collagen-Induced Thrombus Formation

Collagen-induced in vitro thrombus formation in the blood of patients with diabetes was approximately 20% higher than in healthy people; although, it was statistically nonsignificant (Figure 1). Resveratrol significantly reduced thrombus formation by 50% and 60%, respectively, in the blood of healthy people and diabetic patients, while leaving mainly single cells attached to the collagen (Figure 1).

Heparinized blood obtained from healthy volunteers and patients with type 2 diabetes, incubated in the presence and absence of resveratrol and stained with 5 µM DiOC_6_, was perfused over collagen-coated capillaries (VitrotubesTM) (50 µg/mL) at 1000/s for 2 min and visualized by fluorescence microscopy.

### 3.4. Effect of Resveratrol on Collagen-Induced Platelet Function and TXA_2_ Release

Platelet adhesion to the collagen (5 µg/mL) under static conditions was similar in the platelets from the healthy people and patients with diabetes (Figure 2A). Resveratrol reduced adhesion by 85% in the platelets of healthy people and by 55% in diabetic platelets (Figure 2A).

On the other hand, diabetes caused a significant 17% increase in collagen-induced platelet aggregation (Figure 2B). Resveratrol reduced the aggregation by 54% and 52%, respectively, in the platelets of healthy people and patients with diabetes (Figure 2B).

Similar to collagen-induced aggregation, diabetes also increased platelet TXA_2_ production by 47% (Figure 2C). Resveratrol significantly reduced the TXA_2_ production by 38% in platelets from healthy people (Figure 2C). The inhibitory effect of resveratrol on diabetic platelets was two times higher compared with the healthy ones (Figure 2C).

### 3.5. Effect of Resveratrol on Platelets Viability and Anti-Oxidant Capacity

To assess the influence of resveratrol on the viability and survival of blood platelets, we used an MTT test and lactic dehydrogenase (LDH) release in platelet supernatants (Figure 3A,B). Resveratrol (0.01–1.0 mM/L) did not have any influence on LDH release between platelets exposed to resveratrol or control conditions (Figure 3A). Simultaneously, in the same conditions, there was no difference in platelet viability in the MTT test (Figure 3B). Therefore, resveratrol altered neither the viability nor the survival of the platelets in the analyzed samples (Figure 3A,B). Resveratrol also did not influence both the reduced and total gluthatione content in the collagen-activated blood platelets of healthy people and diabetic patients (Figure 3C). On the other hand, resveratrol significantly reduced TBARS accumulation in the collagen-activated platelets of both analyzed groups (Figure 3D).

## 4. Discussion

Increased levels of plasma glucose, blood HbA1c, and fructosamine in the patients with type 2 diabetes selected for the study indicates both the short- and long-term exposition of blood platelets and endothelium to hyperglycemia (Table 1) [3,8]. Therefore, increased platelet aggregation, TXA_2_ production, TBARS accumulation, or elevated activities of hexokinase and G-6-P dehydrogenase may reflect the megakaryocytes and blood platelet metabolic adaptive up-regulation of enzymes responsible for glucose utilization, possibly via an insulin-independent Glut3 transporter, leading to platelet hyperactivation observed in hyperglycemia (Table 1 and Table 2, Figure 1, Figure 2 and Figure 3). The presented data are also compatible with those demonstrating that hyperglycemia induced elevations of the activities of several enzymes responsible for glucose transport, its utilization, and the metabolism of acetyl-CoA leading to hyperactivation of blood platelets in diabetes [3,6,8].

According to the American Diabetes Association (ADA), the recommended glycemic target in patients with diabetes without comorbid conditions and previous hypoglycemia incidents should be below 7% for HbA1c and below 130 mg/dl for preprandial glucose. Therefore, the patients were close to the ADA guidelines (Table 1). Thus, our data indicate correct short- and long-term control of glycaemia in our group of patients (Table 1). Consequently, our results showing only a slight increase in thrombus formation or similar platelet ATP release and gluthatione levels in the patients with diabetes compared with healthy people are compatible with reports in which well-controlled diabetes resulted from the lower risk of vascular complications (Figure 1, Figure 2, and Figure 3C; Table 2) [32,33]. On the other hand, our finding that platelet TXA_2_ production, TBARS accumulation, and collagen-induced aggregation were significantly higher in the platelets of patients with diabetes compared with healthy people may suggest that moderate but chronic hyperglycemia is still a key factor associated with platelet hyperactivity (Figure 2 and Figure 3, Table 1).

In this study, the effects of resveratrol (Res) on platelet metabolism and function in in vitro conditions were investigated. The presented data, showing a significant reduction of thrombus formation by Res in the blood of healthy donors and patients with diabetes, confirms its anti-thrombotic action (Figure 1) [18,19,34]. The fact that the resveratrol’s effect was more pronounced in the blood of patients with diabetes may suggest its important role in reducing thrombotic complications in patients with hyperglycemia (Figure 1). Moreover, the observation that Res reduced both platelet collagen-induced adhesion and aggregation revealed that both the early and late phase of platelet activation is down-regulated by this compound (Figure 1 and Figure 2). This is compatible with previous reports which showed that resveratrol reduced thrombin-induced aggregation by the inhibition of platelet Ca^2+^ signaling [19].

The findings that resveratrol (Res) also inhibited the activities of platelet hexokinase, G-6-P-dehydrogenase, aconitase, and isocitrate dehydrogenase indicate that the compound reduced platelet activation and function by the inhibition of glucose metabolism and, consequently, energy production (Figure 1 and Figure 2, Table 2) [12,13,15,17]. Moreover, the fact that resveratrol reduced platelet ATP release and TXA_2_ production suggests that the platelet down-regulation by this compound is related to the reduction in the release and production of the secondary platelet mediators (Figure 2, Table 2). Additionally, resveratrol-induced reduction of TXA_2_ production is compatible with data showing that, like aspirin, resveratrol significantly inhibits cyclooxygenase-1 activity leading to platelet inactivation and the reduction of thrombotic events (Figure 2) [35]. This anti-platelet property may make resveratrol useful for reducing the incidence of vascular complications, especially in patients with aspirin resistance (Figure 2C) [21]. Furthermore, TBARS accumulation was reduced by resveratrol in the platelets of healthy people and patients with diabetes, which confirms its effectiveness in the reduction of lipid peroxidation (Figure 3C) [36,37]. Surprisingly, resveratrol, as a natural anti-oxidant, did not influence GSH levels in the platelets of both the analyzed groups (Figure 3). This may be explained by the resveratrol-induced reduction of activity of platelet G-6-PDH, the enzyme which is the first step of the pentose phosphate pathway and the major source of NADPH for gluthatione reduction (Figure 3C).

The fact that resveratrol-related reduction of thrombus formation and TXA_2_ production was more pronounced in patients with diabetes suggests that resveratrol may reduce thrombotic complications by the reduction of TXA_2_ production in hyperglycemia (Figure 1 and Figure 2). Moreover, the observation that the activities of aconitase and isocitrate dehydrogenase were more reduced by resveratrol in diabetic platelets compared with platelets of healthy people, indicates that, in hyperglycemia, resveratrol also reduced platelet function by the reduction of oxidative metabolism and energy production (Table 1 and Table 2, Figure 1 and Figure 2).

In our studies, we used high concentration of resveratrol, which gave us the reduction in thrombus formation and platelet metabolism. However, there was no suppression in the MTT test and no release of LDH to the extra-platelet compartment, which confirms that the effect of resveratrol on platelet metabolism and function was not related to its toxicity (Figure 3).

The resveratrol effect was more potent on platelet function in patients with diabetes; therefore, it may suggest that such a compound may be beneficial in this group of patients. There are a number of studies demonstrating that in in vitro conditions resveratrol is a chemoprotective agent in cancer, cardiovascular disease, inflammation, or diabetes. However, the effects of resveratrol in vivo studies are less pronounced, possibly due to its low bioavailability [9,10,11,12,13,14,15,16,17,18,19,20,21,38].

In our studies, we used trans-resveratrol, at a concentration of 250 µM (50 µg/mL), directly incubated for 30 min with whole blood and blood platelets. In such conditions, resveratrol statistically reduced thrombus formation, platelet function, and metabolism. At the same time, it did not induce any toxicity (Figure 1, Figure 2 and Figure 3, Table 2).

There are also results from in vivo studies on the effect of resveratrol in patients with diabetes. The clinical data showed that resveratrol at a wide range of concentrations (10 mg to 14 g daily) reduced fasting glucose, HBA1c, or increased insulin sensitivity, but the effect was observed after several weeks or months of the supplementation and was without evident adverse effects [38]. Therefore, this indicates that it is the transport, the accumulation in tissue, metabolism, deposition, and receptor binding that matters in the bioavailability and effectiveness of resveratrol. There is also a type of derivative that may affect resveratrol bioavailability [38].

Standard oral resveratrol supplementation, recommended to buy over the counter, is usually 100–2000 mg/d. There are, however, some issues of resveratrol bioavailability. Some studies revealed that a single oral dose of resveratrol (25–5000 mg) correlated with its mean peak plasma levels in healthy people, and resveratrol at 5000 mg gave plasma levels about 500 ng/mL [17,18,19,20,21]. Therefore, in our studies, we used resveratrol 50 µg/mL at a much higher dose compared with standard single oral supplementation. On the other hand, some data showed that when radioactive resveratrol was used in humans, the urine secretion of resveratrol, in the form of metabolites, was detected as much as 25–50% [38]. Therefore, it shows that the absorption of resveratrol from the gastrointestinal track is higher than generally expected [38]. In conclusion, all the data confirm more research is required to assess the bioavailability of resveratrol and other polyphenols in humans.

However, in our studies, we focused on finding that an exposition of blood to resveratrol may have an anti-thrombotic effect, regulate glucose metabolism, and affect TXA_2_ production both in healthy and diabetic people. Therefore, our findings are only a tiny contribution to a wide discussion on the effectiveness and safety of resveratrol supplementation in healthy people and patients with diabetes. More research is required in the future to investigate the effect of the oral/intravenous supplementation of resveratrol on reducing thrombotic complications in patients with diabetes, but also to investigate any possible adverse effects, such as bleeding problems.

## 5. Conclusions

Increased platelet aggregation, TXA_2_ production, and TBARS accumulation in patients with type 2 diabetes may be due to elevated activities of key enzymes responsible for glucose utilization in chronic hyperglycemia.

Resveratrol reduced platelet adhesion, aggregation, and thrombus formation. The fact that this compound also inhibited the activities of platelet hexokinase, G-6-P-dehydrogenase, aconitase, and isocitrate dehydrogenase indicates that resveratrol reduced platelet function by the inhibition of glucose metabolism and energy production. Moreover, resveratrol reduced platelet ATP release and TXA_2_ production, which may suggest that platelet down-regulation by resveratrol is related to the reduction in the release and production of the secondary platelet mediators.

Resveratrol-related reduction of platelet function and TXA_2_ production, or the inhibition of aconitase and isocitrate dehydrogenase activities, were more pronounced in the blood of patients exposed to hyperglycemia. Therefore, resveratrol may be useful in controlling platelet hyperactivation in type 2 diabetes.

## Figures and Tables

**Figure 1 nutrients-14-01633-f001:**
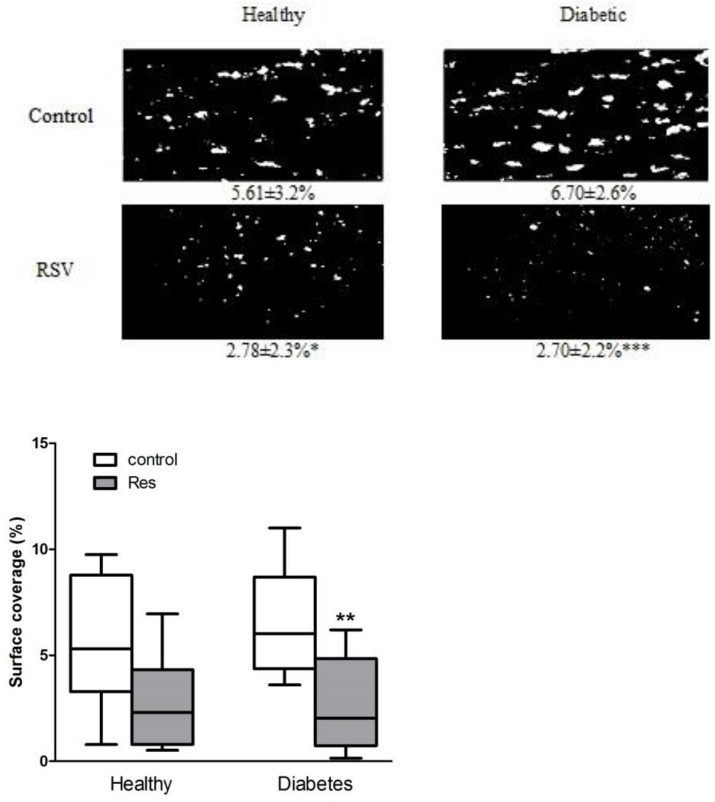
Resveratrol reduced collagen-induced thrombus formation in healthy and diabetic patients in in vitro flow studies. Data are presented as surface coverage in percentage (%) and are the means ± SD or whiskers: min to max from 8 to 10 observations. Significant differences from respective control data without trans-resveratrol (RSV 0.25 mM), (paired Student’s test * *p* < 0.05, ** *p* < 0.01, *** *p* < 0.001). Res: resveratrol.

**Figure 2 nutrients-14-01633-f002:**
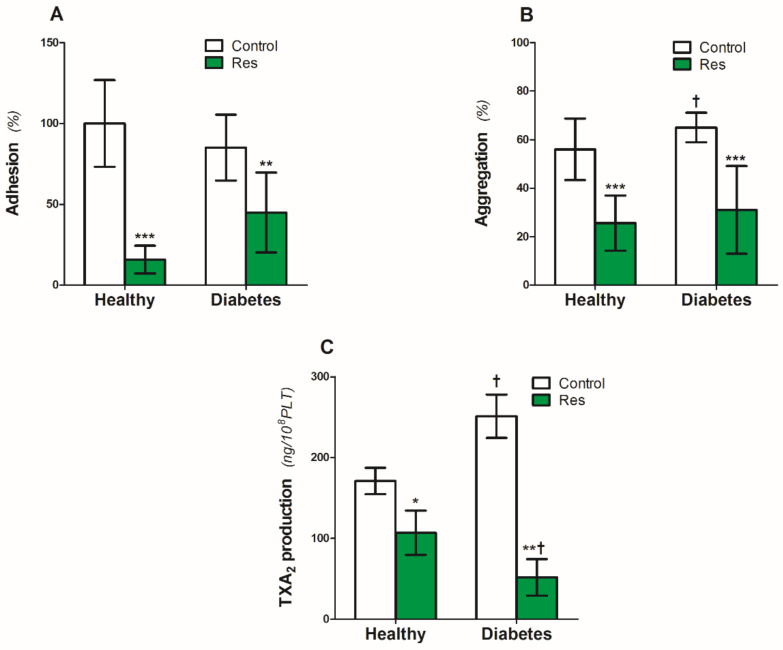
Effect of resveratrol on the blood platelets function of healthy people and patients with diabetes. (**A**): Platelet adhesion, (**B**): Collagen-induced aggregation, (**C**): TXA_2_. Data are the means ± SD from five experiments performed in duplicate. Significant effect of: trans resveratrol (* *p* < 0.05, ** *p* < 0.005, *** *p* < 0.001) (paired *t*-test) and diabetes († *p* < 0.05) (unpaired *t*-test). Abbreviations: Res: trans-resveratrol 0.25 mmol/L.

**Figure 3 nutrients-14-01633-f003:**
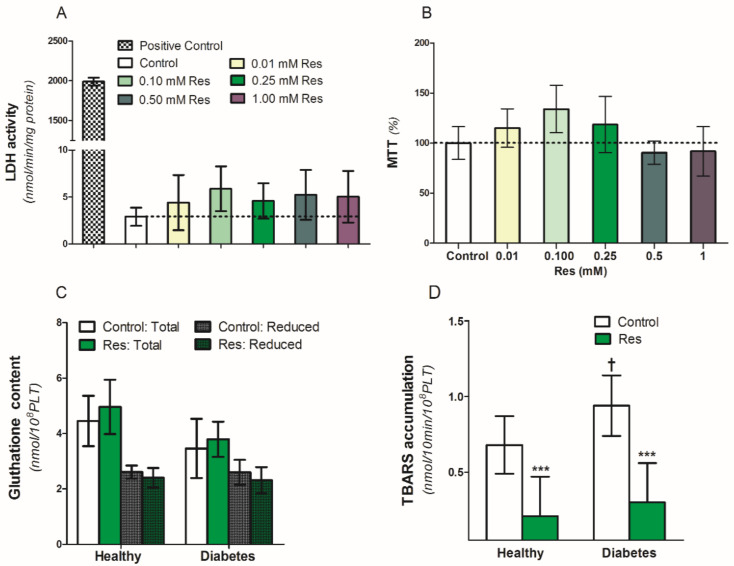
Effect of resveratrol on blood platelets viability and oxidation parameters of healthy people and patients with diabetes. (**A**): LDH activity in supernatants, (**B**): MTT assay, (**C**): Gluthatione content, (**D**): TBARS accumulation. Data are the means ± SD from five experiments performed in duplicate. Significant effect of: trans-resveratrol 0.25 mmol/L (*** *p* < 0.005) (paired *t*-test) and diabetes († *p* < 0.05) (unpaired *t*-test). Abbreviations: Res: trans-resveratrol 0.25 mmol/L for (**C**,**D**).

**Table 1 nutrients-14-01633-t001:** Basic laboratory parameters in healthy donors and patients with diabetes.

Data	Healthy Donors	Patients with Diabetes
No of subjects	8	10
Men/Women	3/5	4/6
Duration of the disease (years)	NA	5.3 ± 3.3
Age (years)	59.4 ± 10	55.4 ± 9.6
Plasma glucose (mg/dL)	93 ± 5	130 ± 34 ^††^
Plasma fructosamine (μmol/L)	230 ± 24	320 ±29 ^†††^
Hemoglobin A1c (%)	5.1 ± 0.5	7.1 ± 1.0 ^†††^

Data are the means ±SD from 8 to 10 observations for laboratory parameters. Significant effect of diabetes: (^††^
*p* < 0.01^, †††^
*p* < 0.001) (Student’s *t*-test). NA: Non-applicable.

**Table 2 nutrients-14-01633-t002:** Effect of resveratrol on selected parameters of energy metabolism in blood platelets of healthy people and patients with diabetes.

Parameter/Conditions	Healthy Donors	Patients with Diabetes
	Hexokinase activity (nmol/min/mg)	
Control	53.1 ± 10.7	70.0 ± 6.49 ^†^
Res	27.1 ± 8.58 **	42.8 ± 12.0 **^†^
	G-6-P-Dehydrogenase activity (nmol/min/mg)	
Control	88.2 ± 13.3	110 ± 9.68 ^†^
Res	44.1 ± 19.7 **	41.8 ± 29.3 **
	Aconitase activity (nmol/mg)	
Control	6.8 ± 0.8	8.4 ± 2.3
Res	3.5 ± 1.8 *	1.6 ± 1.3 **^†^
	Isocitrate dehydrogenase (nmol/min/mg)	
Control	29.7 ± 11.5	30.5 ± 11.8
Res	15.0 ± 4.72 *	10.0 ± 6.35 **^†^
	Platelet ATP release (nmol/mg)	
Control	8.1 ± 5.3	7.2 ± 3.9
Res	2.9 ± 1.8 *	3.3 ± 2.8 *^†^

Data are the means ± SD from five experiments performed in duplicate. Significant effect of: trans-resveratrol (* *p* < 0.05, ** *p* < 0.005) (paired *t*-test) and diabetes (^†^
*p* < 0.05)(unpaired *t*-test). Abbreviations: Res: trans-resveratrol 0.25 mmol/L.

## Data Availability

The data that support the findings of this study are available from the corresponding author upon reasonable request.

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
