# Peer review of "Resveratrol Inhibits Metabolism and Affects Blood Platelet Function in Type 2 Diabetes"

_nutrients, 2022, doi:10.3390/nu14081633_

Round 1

Reviewer 1 Report

With this work, the authors intend to demonstrate that resveratrol has a protective role in relation to the effects of hyperglycemia, characteristic of diabetes, in particular on blood platelet activity and thrombus formation.

In general, an extensive study was made of various parameters that ensure the quality of the results obtained. It is written, in clear English with few flaws.

I have just a few questions on the results and their interpretation.

Please define the positive controls used. Authors suggest their use but do not describe  them in every experiement.

In the presentation of the results, when percentages of differences between conditions are presented I would have liked to see, not only the differences of the mean but also the SD for each case.

in the section: Effect of resveratrol on platelets viability and antioxidant capacity, I think that it would be important to refer to the maximum total time of the experiment.

Discussion:

Line 299: this paragraph needed a better interconnection with the literature.

line 321: I could not agree with this sentence since it would be necessary to evaluate diabetic patients with controlled glycemic levels to make this conclusion. 

Given that most of the results obtained by the authors show that the effects of resveratrol on platelet function are similar in both healthy and diabetic , I think that the effect on healthy humans should be discussed further, including hypothesizing about its consumption in terms of prevention, but also potential adverse effects such as bleeding.

I also suggest a better discussion regarding the doses used and those that can be found in the diet and an explanation for the use of the dose in question in the article.

Author Response

Dear Reviewer,

Please find my response to your comments in the attachment below.

Kind regards,

Anna

Reviewer 2 Report

Dear Editors and Authors, I have carefully read the manuscript "Resveratrol inhibits metabolism and affects blood platelet function in type 2 diabetes". The paper is rather interesting and well written.

However, I have a couple of major comments on this manuscript

Major comments

  1. Authors state the following (lines 214-218):

“Blood HbA1c in healthy people was below the recommended cut-off value for non-diabetic people, and in the diabetic patients was above the general diagnostic criteria for diabetes according to the Polish Diabetes Association followed by the American Diabetes Association (Table 1).”

At the same time the data say that glycated hemoglobin level was:

“<5.9% - the reference value given by the manufacturer.”          

Authors should provide the real value to see if there were any significant differences.

  1. It would be more interesting and informative to have the results obtained with the orally treatment of patients. It is well known that polyphenols have very low digestibility. The results obtained in vitro may be not so evident as in vivo and in the clinic.  Can you provide the results or to comment?

Author Response

Reviewer 2

Dear Editors and Authors, I have carefully read the manuscript "Resveratrol inhibits metabolism and affects blood platelet function in type 2 diabetes". The paper is rather interesting and well written.

However, I have a couple of major comments on this manuscript

Major comments

  1. Authors state the following (lines 214-218):

“Blood HbA1c in healthy people was below the recommended cut-off value for non-diabetic people, and in the diabetic patients was above the general diagnostic criteria for diabetes according to the Polish Diabetes Association followed by the American Diabetes Association (Table 1).”

At the same time the data say that glycated hemoglobin level was:

“<5.9% - the reference value given by the manufacturer.”          

Authors should provide the real value to see if there were any significant differences.

My answer:

The real value for HbA1c has been provided and is in the table 1.

  1. It would be more interesting and informative to have the results obtained with the orally treatment of patients. It is well known that polyphenols have very low digestibility. The results obtained in vitro may be not so evident as in vivo and in the clinic.  Can you provide the results or to comment?

My answer:

Indeed, the data showing the effect of the oral treatment with resveratrol would be more informative than in vitro studies. So there are some limitations of our studies and in the future it would be worth investigating more extensively the effect of oral supplementation of resveratrol/polyphenols in patients with diabetes.

As in our studies we only checked the effect of resveratrol on thrombus formation and blood platelets in in vitro conditions so unfortunately, we cannot provide any in vivo data at this moment.

There are numbers of studies demonstrating that in in vitro conditions resveratrol is a chemioprotective agent in cancer, cardiovascular disease, inflammation or diabetes however the effects of resveratrol in vivo studies are less pronounced possibly due to its low bioavailability.

In our studies we used trans-resveratrol at concentration 250µM (50µg/mL) directly incubated for 30 min with whole blood and blood platelets. In such conditions resveratrol statistically reduced thrombus formation, platelet function and metabolism, at the same time, it did not induce any toxicity (Figure 1, 2, 3, Table 2).

There are also results from in vivo studies on the effect of resveratrol in patients with diabetes. The clinical data showed that resveratrol at a wide range of concentrations (10mg to 14g daily) reduced fasting glucose, HBA1c or increased insulin sensitivity but the effect was observed after several weeks or months of the supplementation and without evident adverse effects (Walle, 2011).  So it indicates that it is the transport, the accumulation in tissue, metabolism, deposition, receptor binding that matters in the bioavailability and effectiveness of resveratrol. There is also a type of derivatives that may affect resveratrol bioavailability (Walle, 2011).

Standard oral resveratrol supplementation recommended to buy over-the-counter is usually 100-2000mg/d). There are, however, some issues of resveratrol bioavailability.  Some studies revealed that single oral dose of resveratrol (25-5000mg) was correlated with its mean peak plasma levels in healthy people, and resveratrol at 5000mg gave plasma levels about 500ng/mL (Boocock et al. 2007). So in our studies we used resveratrol 50µg/mL at much higher dose compared to standard single oral supplementation. On the other hand, some data showed that when radioactive resveratrol was used in humans, the urine secretion of resveratrol in forms of metabolites was detected as much as 25-50% (Walle, 2011). So it shows that absorption of resveratrol from the gastrointestinal track is higher than generally expected (Walle, 2011). In conclusion, all the data confirm more research is required to assess bioavailability of resveratrol and other polyphenols in humans.

But in our studies we focused on finding that an exposition of blood to resveratrol may have antithrombotic effect, regulate glucose metabolism and affect TXA2 production both in healthy and diabetic people.  So our findings are only a tiny contribution to a wide discussion on the effectiveness and safety of resveratrol supplementation in healthy people and patients with diabetes. Definitely, more research is required in the future to investigate the effect of oral/intravenous supplementation of resveratrol on reducing thrombotic complications in patients with diabetes but also to investigate any possible adverse effects such as bleeding problems.

Round 2

Reviewer 1 Report

The revision made by the authors was sufficiently robust and clarified my doubts regarding the work previously presented.

Reviewer 2 Report

There are several typos in the text of the conclusion.
In general, the article has become better and the authors answered the questions posed.